# Combination prevention package of interventions for reducing vulnerability to HIV among adolescent girls and young women in Nigeria: An action research

Olujide Arije[1]*, Ekerette Udoh[2], Kayode Ijadunola[3], Olusegun Afolabi[3], Joshua Aransiola[4], Godpower Omoregie[2], Oyebukola Tomori-Adeleye[2], Obiarairiuku Ukeme-Edet[2], Oluwole Fajemisin[2], Rachel Titus[1], Adedeji Onayade[1,3]

1 Institute of Public Health, College of Health Sciences, Obafemi Awolowo University, Ile-Ife, Nigeria, 2 Society for Family Health, Abuja, Nigeria, 3 Dept. of Community Health, Obafemi Awolowo University, Ile-Ife, Nigeria, 4 Dept. of Sociology, Obafemi Awolowo University, Ile-Ife, Nigeria

* olujidearije@oauife.edu.ng

**Data Availability Statement:** All relevant data are within the paper and its Supporting Information files.

## Abstract

### Background

Adolescent girls and young women (AGYW) in Nigeria are especially at risk of HIV in Nigeria. Their vulnerability to HIV is linked to multiple concurrent sexual relationships, transgenerational sex, and transactional sex, amongst other factors. These factors have sociocultural contexts that vary across a multi-cultural country like Nigeria. The aim of this study was to use an innovative collaborative approach to develop a minimum HIV prevention package for AGYW which is responsive to sociocultural settings and based on combination HIV prevention.

### Methods

We conducted action research to develop and implement actionable HIV prevention intervention models that address AGYW's vulnerabilities to HIV in three Nigerian States and the Federal Capital Territory (FCT) Abuja. The action research adopted the breakthrough series (BTS) collaborative, which accelerates improvement through mutual learning. The BTS implementation involved rapid Plan-Do-Study-Act (PDSA) cycles: an iterative process to plan and implement a basket of interventions. Problems or problematic situations, termed change topics, for which interventions could be carried out were identified in each study location. Using participatory approaches during a series of meetings called learning sessions, specific and innovative interventions, termed change ideas, were developed. These learning sessions were conducted with young women groups and other stakeholders. The change ideas were tested, studied, adapted, adopted, or discarded at each participating site. Exposure to and uptake of the implemented interventions was assessed in the study areas using a household survey with 4308 respondents, 53 focus group discussions, and 40 one-on-one interviews in intervention and control study sites.

**Funding:** This research was funded by the Global Fund to Fight AIDS, Tuberculosis and Malaria as part of its HIV New Funding Model. Olujide Arije was supported by the African Academy of Sciences (AAS) under a DELTAS Africa Initiative grant [107768/Z/15/Z] as part of the Consortium for Advanced Research Training in Africa (CARTA). CARTA is jointly funded by DELTAS, the Carnegie Corporation of New York [B 8606.R02], and Swedish International Development Cooperation Agency (SIDA) [54100029]. The funders had no role in study design, data collection and analysis, decision to publish, or preparation of the manuscript.

**Competing interests:** The authors have declared that no competing interests exist.

## Results

Five categories of interventions were collaboratively developed, namely: Parental communication; Peer to peer interventions; Facilitator-led interventions; Non-traditional outlets for condoms, and Social media-based interventions. A good reach of the interventions was demonstrated as 77.5% of respondents reported exposure to at least one type of intervention. Nearly half of the respondents reported being exposed to the parental communication interventions, while 45.1% reported being exposed to the youth facilitator-driven interventions. Social media interventions had the lowest penetration. Also, there was between 15 to 20 positive percentage point difference between intervention and control for the uptake of HIV testing, and between 5 to 9 positive percentage point difference for uptake of male condoms. These differences were statistically significant at p<0.001.

## Conclusions

Interventions developed through participatory approaches with young people and well-tailored to local realities can improve the acceptability and accessibility of programs that are able to reduce the risk of HIV infection among AGYW.

## Introduction

Adolescents constitute about 7% of the total number of individuals with HIV in Nigeria [1], and young women are disproportionately affected by HIV compared to their male counterparts [2]. HIV incidence per 1,000 uninfected populations among male adolescents aged 10–19 years was 0.53 compared to 1.3 among females [3]. Also, females aged 20–24 years had nearly four times the prevalence of HIV compared with males of the same age group (1.3% vs. 0.4%) [4]. The drivers of the HIV epidemic among adolescents and young people in Nigeria include low personal risk perception, multiple concurrent sexual partnerships, and transactional and inter-generational sex [2, 5, 6]. Entrenched gender inequalities and inequities, chronic and debilitating poverty, and persistence of HIV and AIDS-related stigma and discrimination also significantly contribute to the spread of the infection [2].

The complex nature of the determinants of HIV among adolescent girls and young women (AGYW) requires intervention approaches that have a clear understanding of the disease's epidemiology. Despite the myriad programs and agencies offering HIV-related interventions in the country, the level of knowledge of the infection among young people, their uptake of counselling and testing services, and access to other prevention and care services remain inadequate [7]. Some national programs developed specifically for young people include the Family Life HIV/AIDS Education (FLHE) Curriculum for Junior Secondary School in Nigeria, an abstinence-only curriculum [8, 9], and the National Youth Service Corps peer education program for in-school youth [10]. Some of the gaps in these programs are that the young people were not adequately involved in developing, implementing, and evaluating the programs [7].

Ordinarily, interventions that address the determinants of HIV infection among AGYW ought to be grounded in the context of their vulnerabilities to HIV while proffering accessible and acceptable solutions with their participation. The action research methodology can help in the development of such contextual interventions. It offers a platform that allows young persons and other stakeholders to jointly identify a relevant problem, act together to solve it, cooperatively review to assess their effort's success, and attempt other solutions if the present

one was unsatisfactory [11]. Reason and Bradbury [12] define action research as
"...*participatory, democratic process concerned with developing practical knowing in the pursuit of worthwhile human purposes, grounded in a participatory worldview*". It requires an active and iterative collaboration of researchers and participants in its design, implementation, and evaluation. It offers an opportunity to develop tailored, innovative and adaptive HIV prevention solutions for those affected by the problem.

Action research has been used in different contexts in Nigeria. In a study among rural youths in Nigeria, action research was used to target the youths and their communities with HIV prevention programming that addressed the gendered nature of HIV vulnerability [13]. The researchers focused on developing communities' HIV/AIDS competence and school-based HIV interventions to improve a school-based sexual and reproductive health program. Fakoya et al. [14] reported on the A360 intervention which integrated a human-centered design and youth-led participatory action research as an innovative and replicable approach to reducing unmet need for contraception among adolescents and young people (AYP). The four phases of the intervention included: youth-engaged formative research; collaborative analysis to generate themes to inform intervention design; prototyping of interventions; and adaptive (ongoing refinement, critical reflection, and iterative evaluation of solutions) implementation. Important lessons from this study for using action research in global health research included forming transdisciplinary teams, centering empathy by using methodologies that amplify the voice of participants e.g. qualitative data collection, purposive selection of participants, rapid prototyping of solutions, and having tangible services or products. Action research intervention involves cycles of dialogue and action among stakeholders in the area of interest, and it is a valuable approach for quality improvement in health programming. Complex, persistent or unstructured problems cannot be tackled effectively by the more traditional research approaches that do not adequately address the problems underlying social, political, economic, cultural, and ethical aspects [15].

The Breakthrough Series (BTS) collaborative lends itself to the ideals of action research [16. 17]. Breakthrough Series (BTS) is a learning collaborative (LC) approach that utilizes a quality improvement method designed to enable participating teams to make dramatic improvements in a focused practice topic over a short period [17]. It is an improvement approach that relies on spreading and adapting existing knowledge to multiple settings simultaneously [18]. The BTS Collaborative methodology was developed in 1995 by the Institute for Healthcare Improvement (IHI) and Associates in Process Improvement (API) [17]. The BTS collaborative allows researchers and potential research beneficiaries to use existing and available interventions to build custom-made solutions to identify local problems/challenges, test the solutions on a small scale, and rapidly assess their viability.

The rapidity of the BTS Collaborative methodology allows for the development of several solutions while testing them on a small scale to identify the one with the most significant potential for success at scale. More so, learning collaborative is not intended to create an entirely new body of knowledge but provide what might be the missing link between best practice and actual practice [19]. The method has been used extensively in health care delivery in high-income societies and much less so in low and middle-income settings [20, 21]. The BTS collaborative approach has been implemented in Nigeria, but mostly in clinical settings. For example, it was used for providing a sustainable framework for the role of community health workers in promoting retention in HIV care [22], improving prevention of mother to child transmission of HIV (PMTCT) [23, 24], and improving childhood immunization rate [25].

Combination prevention is the recommended approach for comprehensive prevention of HIV. According to the UNAIDS Prevention Reference Group [26], combination HIV prevention is defined as "*The strategic, simultaneous use of different classes of prevention activities*

*(biomedical, behavioral, structural) that operate on multiple levels (individual, community and societal/structural), to respond to the specific needs of particular audiences and modes of HIV transmission, and to make efficient use of resources through prioritizing partnership, and engagement of affected communities.*" The Nigerian National HIV/AIDS Prevention Plan (NPP) 2010–2012 [27] introduced the combination prevention approach as Minimum Prevention Package of Interventions (MPPI) to scale-up evidence-based programming using targeted interventions and standardized intervention packages at scale. Strategies under the behavioral components include outreach, peer education, and condom and lubricant programming. Strategies under the biochemical component include HIV counseling and testing (HCT), prevention of mother-to-child transmission (PMTCT), and sexually transmitted infection (STI) control and treatment. The structural component (which address gender issues, stigma and discrimination, policy issues, and individual empowerment) includes community mobilization and dialogue (empowerment and capacity building), advocacy, and individual empowerment/income-generating activities. The aim of this study was to use an innovative collaborative approach to develop a minimum HIV prevention package for AGYW which is responsive to sociocultural settings and based on combination HIV prevention. The country is still in the process of developing a scalable guide for the implementation of community-based HIV programs focused on AYP in Nigeria (personal communication). The package of care put forth in the Nigerian NPP 2010–2012 and the MPPI are generic while the prevention package we present in this study are adaptive and contextualized. This paper describes the intervention component and outcomes of the action research used to develop the package of prevention.

## Conceptual framework

To provide a conceptualization for how multilevel (structural, behavioral and biological) interventions reduce HIV incidence, the mechanisms at play, and the implementation outcomes assessed in our study, we adopted the conceptual framework used by Chimbindi et al. [28] in their intervention to reduce HIV incidence among AGYW in South Africa (Fig 1). In the framework, distal factors such as household and individual sociodemographic factors, and structural interventions interact with proximate factors such as sexual and health behaviors, health and biomedical interventions, and behavioral interventions to yield outcomes that

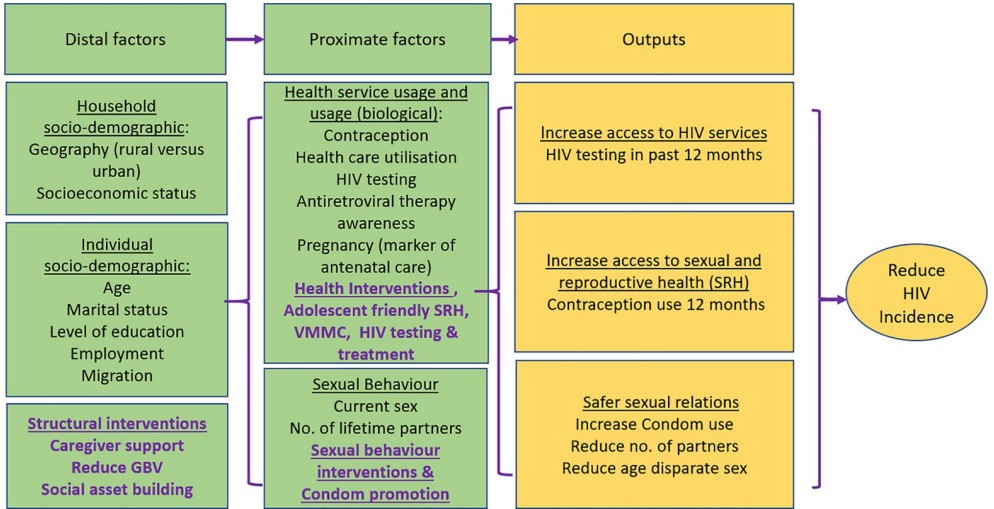

**Fig 1. Conceptual framework for effect of combination prevention on HIV incidence in adolescent girls and young women.** Source: Chimbindi N, et al. PLoS One. 2018;13: 1–17.

include increased access to HIV and SRH services, and reduction in risky sexual behaviors. Ultimately, these lead to reduction in HIV incidence. Our study provides insight into how the adolescent HIV response within the country based on the combination prevention can be particularized at sub-country levels.

## Methods and materials

### Study design

The Institute of Public Health (IPH), Obafemi Awolowo University, Nigeria, collaborated with the Society for Family Health (SFH), a national non-governmental organization, to develop and implement the action research to reduce HIV vulnerability among AGYW in Nigeria from 2016 to 2017. This research has a mixed-method design involving quantitative (cross-sectional descriptive study) and qualitative (focus group discussions with AGYW, and one-on-one interviews with selected key informants) data collection methods. We carried out the study in Akwa-Ibom, Kaduna, Oyo States, and the Federal Capital Territory (FCT Abuja). These states are located in different geopolitical zones, hence, represent the multi-cultural nature of the country. We purposively selected two local government areas (LGA) as implementation sites and one as the control site in each study location (Table 1). The selection of the states was based on states that had the highest HIV prevalence amongst the target population (15–24 years) within their zones. Updated national survey (NAIIS 2018) shows that Akwa Ibom State has the highest HIV prevalence among persons aged 15–49 years in the country at 5.6%. This was 1.3% in the FCT Abuja, 0.9% in Kaduna, and 0.8% in Oyo [29]. The selection of LGAs in which interventions were implemented was based on the estimated youth population of the LGAs (highest proportion of youths by extrapolation from the 2006 Nigerian census (last national census held)), and the absence of any youth-focused HIV prevention interventions in the area at the time of our intervention.

### Implementation approach

The BTS collaborative methodology drove the approach taken in this action research. Our BTS collaborative objective was to develop a comprehensive *'change package.'* We defined a change package as a collection/combination of innovative interventions that have been tested on a small scale and found to give the desired change, therefore, having a good prospect for scale-up. Secondly, the developed change package must align with the Combination HIV Prevention

Table 1. List of study states and LGAs indicating intervention and control LGAs.

| State | LGA |
| --- | --- |
| Akwa-Ibom | Ikot-Ekpene |
| | Oron |
| | Eket (control) |
| FCT | Bwari |
| | Gwagwalada |
| | Abaji (control) |
| Kaduna | Chikun |
| | Lere |
| | Sabon'gari (control) |
| Oyo | Ogbomosho North |
| | Ibadan North |
| | Afijio (control) |

model, the basis for MPPI, with behavioral, biomedical, and structural components. On small scales, we tested the potential intervention models to be included in the change package using the Plan-Do-Study-Act (PDSA) cycle (Fig 2). Desired change was contextual for each change idea implemented and was evaluated using the decision rule for implemented change ideas (further described below).

## BTS collaborative management

Teams were constituted at national, State, and LGA levels to plan and implement the BTS collaborative. The National BTS team comprised principal actors from IPH, SFH, and the National Agency for the Control of AIDS (NACA). Each State BTS team was comprised of IPH and SFH State-level project focal persons and managers, focal persons from the State Agencies for the Control of AIDS (SACA) and the State Ministry of Health (SMoH), and representatives of State-level youth-focused Civil Society Organisations (CSO). The local BTS teams, which operated at the LGA level, comprised of focal persons from the Local Agencies for the Control of AIDS (LACA), male and female community youth leaders, AGYW representatives, and representatives of youth-focused community-based organizations (CBOs). Fig 3 shows the line of communication across the BTS teams. Communication across the BTS collaborative was facilitated by regular meetings at local, state and national levels. The research teams at the local, state and national levels were led by research staff from IPH while the corresponding BTS management teams were led by program staff from SFH.

## Learning session

We developed intervention models by selecting problematic situations/issues derived from an objective assessment of the intervention target population. This objective assessment was a baseline survey that identified vulnerability factors to HIV among AGYW. The findings from this

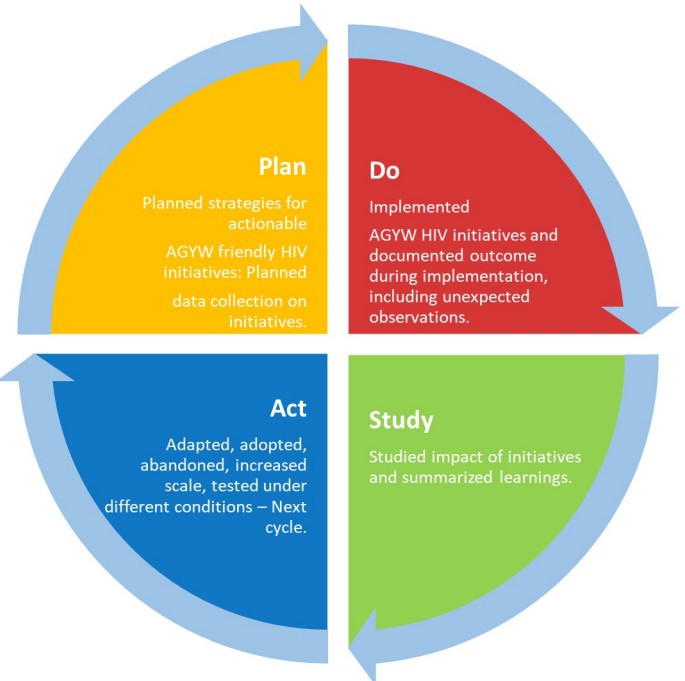

**Fig 2. The Plan-Do-Study-Act (PDSA) cycle.**

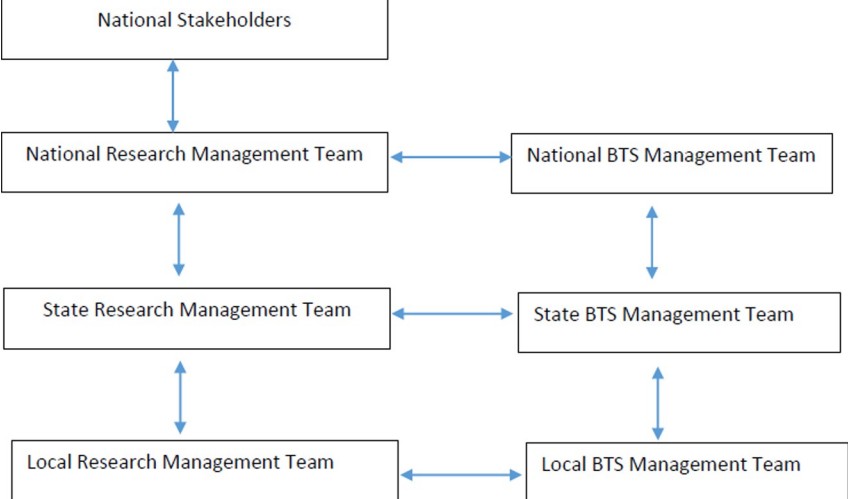

**Fig 3. Organogram and line of communication for the BTS collaborative management.**

baseline survey are published elsewhere [5]. The problematic situations identified were called '*change topics.*' The State BTS teams worked with each of their Local BTS teams to explore the contextual factors unique to the change topics identified in each study location in a series of two-day meetings called learning sessions. We developed a prioritization matrix for deciding on the change topics for which to focus intervention. The prioritization matrix allowed for scoring each change topic on the prevalence of the change topic, perceived public health importance, ease of intervening, and time interval to achieve results if there was an intervention. Scoring of the change topics using the prioritization matrix was conducted during the learning sessions by the local BTS teams (Table 2). The two highest-scoring change topics were selected to be intervened on at the study LGAs each time during two rounds of intervention.

During the learning sessions also, appropriate intervention models (called 'change ideas') were developed to address the prioritized change topics by the State and the LGA teams. We held three sets of learning sessions in all, separately in each study state. The first set of learning sessions across the study locations were strictly for selecting the change topics for intervention and planning the interventions to deploy. In the second learning session, we reviewed interventions implemented so far and planned new change ideas. The last learning session was for reviewing the whole implementation process using the monitoring and evaluation reports from each of the interventions implemented.

**Table 2. Change topic prioritization matrix.**

| Priority Attribute | Indicators | Levels | Score |
|---|---|---|---|
| Impact | Prevalence | ≥50% | 3 |
| | | 21–49% | 2 |
| | | 0–20% | 1 |
| | Public health importance | High | 3 |
| | | Low | 1.5 |
| Feasibility | Ease of intervening | Very easy to implement | 2 |
| | | Not easy to implement | 1 |
| | Time interval for results | Results achievable in <4 weeks | 2 |
| | | Results not achievable in 4 weeks | 1 |

The first learning session was held in each intervention LGA, while the remaining two sessions were held at the State level to allow for cross-pollination of ideas between the LGA teams within a State. Members of the national BTS team participated in the State-level learning session to provide guidance and oversight. The national BTS team also held two national-level review meetings in the course of the project to which some members of the state and local BTS teams were invited to participate. Learning sessions provided fora for sharing knowledge, discussing methodology, and planning action periods (discussed below). The learning sessions corresponded to the 'Plan' and 'Act' phases of the PDSA cycle. During the learning sessions, planning for the implementation of interventions is done. During subsequent learning sessions, the 'Act' phase of the PDSA cycle was included, which entailed deciding whether to abandon, test under different conditions, adapt (with modifications), or adopt (as is). For this, the BTS teams used the decision rule in Table 3 below. Change ideas scored 0 were abandoned, while those scored 1 or 2 were adapted (with modification) or tested under different conditions in the subsequent action period. Those scored 3 were adopted (as is) into the change package.

## Action period

The action period was the time during which change ideas were implemented. This study had two action periods; the first period lasted for three months, and the second lasted for two months. The two were implemented between May and September 2017. Action periods corresponded to the 'Do' and 'Study' phases of the PDSA cycle. New change ideas were started, or previously successful ones were continued or adopted/adapted from another location at the subsequent action period [16, 17]. The 'Study' phase was each change idea's monitoring and evaluation component that captured data based on intervention-specific indicators developed during the planning phase. For instance, the indicators included uptake of services, such as HIV testing service (HTS) and condoms, among those reached with the interventions. Teams identified the successes and challenges they experienced while implementing the change ideas and shared them at the subsequent learning session to enhance knowledge for the entire group. Local community-based organizations were engaged in the implementation of some the change ideas. For example, they provided the youth facilitators for the facilitator-driven interventions (discussed below). CBOs were engaged according to the need in each study site and the research provided site-specific training and support as was necessary for the CBOs that were engaged.

## The change ideas

Each study location had its set of customized interventions (change ideas) that addressed specific vulnerability to HIV. Our assessment showed that the change ideas that had reasonable success could be grouped into five classes: Parental communication, Peer to peer, Youth facilitator-driven; Non-traditional outlets for condoms; and Social media-based interventions. Each of the interventions developed was given names unique names by the respective BTS teams.

**Table 3. Decision rule for implemented change ideas.**

| Score | Operational Definition |
|-------|------------------------|
| 0 | No evidence or suggestion of improvement |
| 1 | Suggestion of improvement but not enough time to meet the test of evidence |
| 2 | Evidence of improvement but not sustained to assess sustainability |
| 3 | Evidence of improvement which has been sustained |

**Parental communication interventions.**   These interventions involved parents in reducing their children's or wards' vulnerability to HIV. It entailed parents' active engagement through established community structures such as women groups in churches or within the community. Parents were trained to become agents of change for their children. In this research, our flagship parental communication intervention in Akwa-Ibom State was called *Item Uwem* that focused on mothers with children in the 15–24-year age group. In FCT Abuja, it was *My Daughter My Pride*, while in Kaduna State, it was *Mother to Daughter (M2D)*.

**Peer to peer communication interventions.**   These were the interventions in which health education, condom distribution, or referral for HIV testing service (HTS) or STI treatment were delivered by a peer who had been engaged as a change agent or peer mentor. In these interventions, the service delivery was often initiated by an intervention facilitator but continued by the peer. The *AYP Cell Meeting* intervention in Akwa Ibom State incorporated peer-led HIV prevention intervention. We had *Girlfriend networking* and *One Mentor Five*, in FCT Abuja, *Tell-a-friend* in Kaduna State, and peer-to-peer condom distribution in Oyo State. These interventions focused on interpersonal communications. The recruitment of AGYW into this type of intervention was by snowballing where recipients were asked to invite their peers.

**Youth facilitator-driven interventions.**   These required a facilitator to deliver services to the recipient throughout the interventions' life span. These interventions generally created platforms for AGYW to learn life and entrepreneurial skills, and receive HIV and sexual and reproductive health (SRH) messages along with condom distribution. The interventions in this category in Akwa Ibom included *Babes Alive* and *AYP Cell Meeting*. The interventions in this category in FCT included *Debunking the Myth and misconception of HIV*, *Skills to Health*, and *Condoms for Us*. The interventions in this category in Kaduna included interventions that target Fathers, Mothers, and Husbands of AGYW and traditional leaders. Meanwhile, in Oyo state, the category's interventions included *Social to Health*, *Interpersonal communication with AGYW*, and *STI/HTS Outreaches*.

**Non-traditional outlets for condom distribution.**   Traditionally, condoms are accessed from pharmacies, drug stores, health facilities, or similar places. These places often present barriers to accessing condoms for AGYW because of stigma, shyness, and cost. This study explored some non-traditional outlets for condoms with some success, such as betting shops, football viewing centres, and barbing/hairdressing salons. There were two such interventions in Akwa Ibom State. In one, condoms were stationed in selected publicly assessed locations (we called these *Stationary condom distributors/dispensers*). The second one had condoms placed with selected young persons who could distribute them freely to their peers on request (*Gallant condom distributors/dispensers*). In Kaduna, the *Condom for Us* intervention incorporated some peer-directed condom distribution. In FCT Abuja, the *Peer-to-peer distribution of condoms* was a flagship non-traditional condom outlet intervention. Also, in FCT Abuja, the *Social to Health* and *STI/HTS Outreach* interventions incorporated some peer-directed condom distribution.

**Social media-based interventions.**   This was carried out in Akwa Ibom and Oyo State only. Specifically, WhatsApp groups were formed to engage AGYW, and those engaged were tasked to recruit other AGYW to join the groups. Online meetings were carried on during which a facilitator creates conversation and passes across SRH and HIV messages. However, this approach was only marginally influential in the higher socioeconomic areas where the AGYW were more likely to afford internet access. Due to the low output in young people joining the groups, the approach was abandoned after initial testing.

## Assessment of exposure to and uptake of the interventions

**Household survey.**   A quantitative household survey was conducted as part of the evaluation of this action research. We used a standard normal deviate at 95% confidence level of

1.96, a precision margin of 5%, a design effect of 1.5, and an estimate of true proportion (10-19-year-olds reporting a history of concurrent multiple sexual partners [6]) of 27%, to determine a minimum sample size of 302. We adjusted this minimum size for non-response to 360 and applied this sample at the LGA level to allow robust data interpretation at the local government level. The total sample size was 4308. A multi-stage sampling technique was employed to sample participants at enumeration area and at household levels.

Respondents were asked if they had been exposed to any sexual and reproductive health intervention in the past six months, the persons who introduced the intervention, what services were introduced, and what services they took up due to the exposure. The instruments were pre-tested in a different population to refine the tools and adapt to the target population's realities without losing the content's validity. The action research design we adopted did not fit a pre-post assessment type of evaluation because the specific interventions implemented were designed as the study progressed. In this study we only report data collected at the end of the intervention The specific outcomes of interest were, if the respondents were reached with an HIV prevention intervention in the past six months, what type of HIV prevention intervention it was, who reached them with the intervention, and if they took up the intervention. Only these outcomes are reported in this current study. The study instruments were interviewer-administered using trained data collectors. The data collectors were trained to describe the local intervention implemented in the different study locations to reduce misclassification bias (see S1 File).

**Focus group discussion (FGDs); In-depth and Key informant interviews (IDI and KII).** In each LGA studied, AGYW were recruited purposively to participate in the FGDs. Also, interviews were held with selected community gatekeepers, health workers, and HIV focal persons. We designed appropriate study guides for the data collection (see S2 File). All discussions and interviews were tape-recorded, transcribed, and translated into English (where local languages were used during sessions).

**Recruitment of research participants.** Before commencement of the household survey, we conducted advocacy visits with community leaders at the selected study locations to gain entry. For each study LGA, a computer-assisted simple random selection of 30 enumeration areas (EAs) was made from a list of all EAs in the LGA. A household listing exercise was done in each selected EA to generate a sampling frame of all eligible households. Eligible respondents were selected from households randomly selected from a household listing exercises.

For the qualitative research participants, eligible respondents were purposively selected based on the various categories of participants. In each LGA, five groups of participants were recruited purposively for FGD sessions including adolescent girls (15–19 years), young women (20–24 years) and young men (18–24 years), AYGW (15–24 years) identified to be at higher risk of contracting HIV due to identified risk practices (including female hawkers around high-risk locations, bar maids etc.), and a group of adult men 30 years or older (because of the role this age group of men tend to play concerning the vulnerability of AGYW to HIV and STI). Each FGD session had 8–10 participants. The research participants included both those were exposed and not exposed to the interventions we carried out. In-depth interviews were held with purposively selected community gate keepers/influencers including at least one religious leader, traditional leader, youth leader, media practitioner, secondary school teacher in each study LGA. In addition, four parents of AYP were interviewed. Key informant interviews were held with selected officials including officers from the primary healthcare department, Women Affairs program, HIV program, and non-profit organizations involved with AYP. All FGD and IDI participants received transportation reimbursement of ₦1000 each ($1≈₦360, 2017) Key informants were not reimbursed because interviews took place in their various offices.

## Data analysis

**Quantitative data.** We conducted tests of the equality of proportions comparing the interventions' uptake among respondents in the intervention and control LGAs. All relevant data are within the manuscript and its Supporting Information files (see S1 Data).

**Qualitative data.** We developed codes from an initial run of a few transcripts. These were refined in a group session and shared among coders who subsequently coded all transcripts from the FGDs, IDI, and KIIs, applying thematic content analysis. A generic codebook was used across all transcripts. Four coders used the ATLAS.ti 8 software for organizing the codes and the units of meaning or concepts they represented. The codes were organized into themes and subthemes. Coding discrepancies were resolved during plenary meetings and final decision when discrepancies remained was made by the lead qualitative researcher on the project (JA). This study reports aspects of the qualitative data relevant to understanding the interventions' dynamics and perceived impact. We viewed acceptability of the interventions as the extent to which our target population considered them appropriate, based on their cognitive and emotional responses to the interventions [30]. On the other hand, we viewed accessibility of the interventions as the relative ease the interventions could be reached in a given location, including the suitability of location, confidentiality/anonymity, and cost of services [31, 32].

## Ethical approval

Ethical approval was obtained for this study from the National Health and Research Ethics Committee of the Federal Ministry of Health, Nigeria with approval number NHREC/01/01/2007-13/09/2016. Written informed consent was obtained from each study participant after an adequate explanation of the study objectives. For participants younger than 18 years, assent to participate in the study was obtained from them after obtaining written informed consent from their parents or guardians.

## Results

To evaluate the reach and uptake of interventions in this study, we interviewed females aged 15 to 24 years old in a household survey. 4308 AGYW were interviewed across the four study States (2868 in the intervention LGAS and 1440 in control LGAs). We also conducted 53 FGDs, 22 IDI and 18 KIIs. For the FGDs, 36 AGYW were recruited in each LGA and categorized into the following groups; adolescent girls (15–19 years), young women (20–24 years), and young men (18–24 years). We had a group of females (15–24 years) identified as at higher risk of contracting HIV due to identified risk practices and their work nature. Such risk factors included those engaging in transactional sex and multiple partnering, females whose occupations put them at risk, including female hawkers around high-risk locations, barmaids, food vendors, and domestic hands. We also conducted FGDs with mothers who had female children in the 15–24-year age group, as well as men in the 30–40-year age group. The latter were included because of the role this age group plays concerning girls and young women's vulnerability to HIV infection and STIs. We held in-depth interviews with 22 selected traditional leaders, religious leaders and youth leaders across the study states. We also held key informant interviews with 18 health workers, and focal persons from the LGA Primary Health Care Department and the State Ministry of Health across the study states.

## Parental communication intervention

In the intervention areas, the proportion of AGYW who indicated that their parents communicated with them on HIV/SRH in the past six months ranged from 23.2% to 40.6% compared

to 3.9% to 18.9% in control LGAs (Fig 4). Among all respondents in the intervention LGAs, HIV testing was the most frequently used service due to exposure to parental communication (19.3%) (Table 4). In comparison, those who reported using HIV testing services following parental communication were fewer in control LGAs (3.5%), and the difference in proportion was statistically significant at p<0.001. The least used service following parental communication was family planning services (2.8%). All services assessed had a higher proportion of uptake in the intervention LGA than the control LGA, and all the differences were statistically significant. Some of the participants of FGD sessions indicated ongoing discussions on reproductive health issues between mothers and their children in the intervention sites. For instance, a 21-year-old young woman said: "... *the woman was talking to the daughter, the daughter poured out her mind, told the mother everything, the mother advised her on some area she needed.*" Another 19-year-old girl similarly said, "*Mothers are now friendly with adolescent girls rather than giving them threatening words, unlike before.*"

Further evidence from other stakeholders in the intervention sites confirmed that some parents were now better informed on giving SRH information to their AGYW wards than before the interventions were carried out in their communities. A female community leader in FCT Abuja reiterated this by saying, "*We now know that we should nurture them. We tell them what they should do since we know that they are mature. If you think that you cannot hold yourself (*i.e., abstinence), *that you cannot maintain yourself, use a condom because of the sickness (HIV)*". A State Ministry of Health HIV desk officer from Kaduna state said: "*Knowledge on HIV has increased. Before our women don't sit with their daughters to talk on sexual issues, but with this program, mothers now discuss with their daughters on sexual issues.*" Similarly, a 39-year-old youth leader in Chikun LGA, Kaduna state said the following:

> "*The benefits of the program are many. One, there is information now. Two, the mothers too are informed, and this will go a long way to help the girl child because every child you see comes from a home, and if the mother is adequately informed, she will also pass the information to her child*".

Parental communication on reproductive health was less evident in control LGAs, as seen in the comment of a respondent who said, "*Some of the parents are afraid of mentoring their*

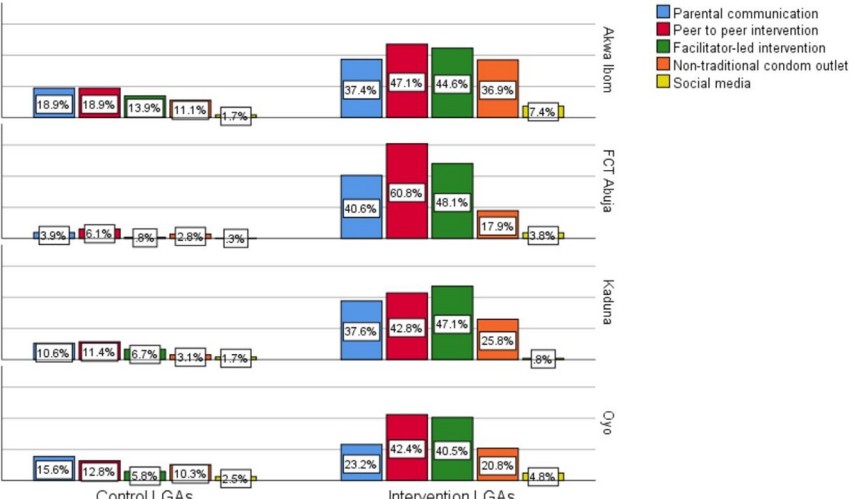

**Fig 4. Respondents reporting exposure to implemented intervention in intervention and control sites across the four study states.**

**Table 4. Uptake of change idea according to exposure to intervention models.**

| Uptake of interventions | Control | Intervention | % Difference | p-value |
|---|---|---|---|---|
| | Freq (%) n = 1440 | Freq (%) n = 2868 | | |
| **Services used following parental communication** | | | | |
| HIV testing | 51 (3.5) | 554 (19.3) | 15.8 | <0.001 |
| STI treatment | 26 (1.8) | 236 (8.2) | 6.4 | <0.001 |
| Family Planning | 12 (0.8) | 80 (2.8) | 2.0 | <0.001 |
| Get condoms | 22 (1.5) | 212 (7.4) | 5.9 | <0.001 |
| **Services used following peer to peer intervention** | | | | |
| HIV testing | 30 (2.1) | 556 (19.4) | 17.3 | <0.001 |
| STI treatment | 6 (0.4) | 125 (4.4) | 4.0 | <0.001 |
| Family Planning | 3 (0.2) | 47 (1.6) | 1.4 | <0.001 |
| Get condoms | 9 (0.6) | 239 (8.3) | 7.7 | <0.001 |
| **Services following facilitator-led intervention** | | | | |
| HIV testing | 39 (2.7) | 647 (22.6) | 19.9 | <0.001 |
| STI treatment | 7 (0.5) | 161 (5.6) | 5.1 | <0.001 |
| Family Planning | 4 (0.3) | 51 (1.8) | 1.5 | <0.001 |
| Get condoms | 5 (0.3) | 263 (9.2) | 8.9 | <0.001 |
| **Uptake of non-traditional condom outlet services** | | | | |
| Collected condoms in the last 6 months at any kind of locations | 16 (1.1) | 348 (12.1) | 11.0 | <0.001 |
| Heard or attended any program were condom was distributed | 36 (2.5) | 955 (33.3) | 30.8 | <0.001 |
| Aware of any condom distribution by any friends or young person | 61 (4.2) | 911 (31.8) | 27.6 | <0.001 |
| Collected free condom from any friends or any other young person | 20 (1.4) | 408 (14.2) | 12.8 | <0.001 |

children. You know our local names for male and female sex organs—It's not good to hear." A participant in the FGD for older men in FCT revealed that parents in Abaji (a control LGA) were generally not favourably disposed to discussing reproductive health issues with their AGYW. He said: "...our people are still looking at it as a taboo for a mother to sit her daughter down or a father sits his son down and tell him this is how to go about a woman, that is to say, the knowledge is still lacking".

## Peer to peer interventions

Across the intervention LGAs, 47.1% to 60.0% of respondents in the household survey reported that they were spoken to by a young person or friend about HIV/AIDS or sex education in the last six months compared to 6.1% to 18.9% across all the control LGAs (Fig 4). Among all respondents in the intervention LGAs, HIV testing was the most frequently reported service used following peer-to-peer communication on SRH (19.4%), followed by getting condoms (8.3%), STI treatment (4.4%) and family planning services (1.6%) (Table 4). All services assessed had a statistically significantly higher proportion of uptake, comparing the intervention LGAs to the control LGAs.

The respondents in the FGD sessions affirmed the intervention programs for AGYW, where adolescents were recruited to give sex education to their peers. For instance, a 19-year-old girl from Oyo State said: "We get information from our friends too. Some young girls were educated, and they also educated many young girls on HIV prevention and treatments so that we don't spread the virus again". Another AGYW participant from Oyo State said: "I introduced my friend to the AGYW program. She, in turn, introduced her friends. Many of my friends attended the program. It was very educative. How I wish it continues". Similarly, a 22-year-old participant from Kaduna State said: "In recent times, there are lots of information we get from

*our friends, about infections contracted through sex. . .".* Stakeholders interviewed affirmed the benefit of the peer-to-peer communication. For instance, a 40-year-old health worker from Kaduna State said: *"Honestly, it has so many benefits, especially the 'Tell-a-friend' as I have said earlier. These AGYW believe in themselves whenever their peers are telling them something; they take it more seriously than when another person is doing that, so "Tell a friend" has helped many of them."*

### Youth facilitator driven interventions

Across the intervention LGAs, 40.5% to 48.1% of respondents in the household survey reported being told about HIV/AIDS in the last six months by a health worker/youth facilitator/ interpersonal communication (IPC) agents compared to 0.8% to 13.9% across all the control LGAs (Fig 4). Among all respondents in the intervention LGAs, HIV testing was the most frequently used service following the exposure to the information (22.6%), followed by getting condoms (9.2%), STI treatment (5.6%), and family planning services (1.8%). Similarly, the respondents in the intervention LGAs had a higher proportion of uptake of all services, and all the differences in proportions were statistically significant (Table 4).

We found some evidence for the reach of the facilitator-led intervention from the FGDs and stakeholder interview sessions. For instance, a 42-year-old mother of an AGYW in FCT Abuja said: *"Yes, some people came to educate the young people in the community on HIV, and many adolescents attended the program."* Similarly, a female adolescent in FCT Abuja said: *"I was personally at the program. They started with a skill acquisition program and also educated us on how to prevent infections and later conducted tests for some of us who agreed to the request".* A 22-year-old female from Akwa Ibom said: *". . .on my street, they gathered some girls in one place and talked to them about this HIV, and most girls that were not aware got to know about it. They talked about ways of preventing it and how to protect yourself as a girl child.".* A 60-year-old community leader in Oyo State said: *"We have some community-based organizations who came to our communities to educate young people on HIV prevention in the last few months. Many of them attended".* Some AGYW participants reported behavioral changes as a result of the intervention. A female participant from Akwa Ibom State said: *". . . (the) benefit is that the project has helped us a lot. . ., it has made us understand that if you have HIV, your life does not end there. It had helped me personally; before I never liked condoms, now I am proud of using it".*

### Non-traditional condom distribution-based interventions

Awareness of where condoms could be obtained, apart from a health facility and chemist/ pharmacy shop, was higher in the intervention LGAs (17.9% to 36.9%) compared with the control LGAs (2.8% to 11.1%) (Fig 4). Among all respondents in the intervention LGAs, 12.1% collected condoms at locations apart from a health facility. Similarly, 33.3% heard of, or attended, a program within their communities where condoms were distributed. Also, 31.8% were aware of condom distribution by friends or youths in their communities, and 14.8% collected free condoms from their friends or youths in their communities in the last six months (Table 4). These proportions were smaller for those reporting similar experiences in control LGAs, and the differences in proportions were statistically significant. According to participants in the FGD sessions, some of the condom outlets they were aware of included barbing salons, sports betting centres, tailoring shops, and sports viewing centres. Some adolescents distributed condoms to their peers. A 20-year-old girl from Akwa-Ibom said: *"Recently when I went to retouch my hair I've seen condoms in the salon, and in the barbing area. So those things are very common".* A 23-year-old male from Akwa Ibom State equally said: *"'My brother showed me the condom he took from the 'Bet Naija centre* (sports betting shop)."

### Social media-based interventions

The reach of the social media approach was shallow. The social media intervention exposure was not very common across all the study sites. As a result of the reported low reach among the respondents, we did not further explore the uptake of services.

## Discussion

In this study, we explored an innovative approach to developing and testing tailored HIV prevention interventions. We successfully developed a basket of intervention models (change package) with reasonable acceptability that can be scaled up to tackle the vulnerability to HIV among AGYW in different settings. These interventions were grounded in the combination HIV prevention approach. Combination HIV prevention continues to be the strategic approach to the national response to HIV/AIDS in Nigeria [33, 34]. The MPPI operationalizes the combination prevention framework in Nigeria, using information about the drivers of the epidemic relating to various target populations, emphasizing dosage and intensity on interventions, and recognizing the processes of behavior change and structural and environmental influencers of behavior [35]. Several authors have successfully implemented HIV prevention interventions based on the MPPI strategy among in-school and out-of-school adolescents, including community mobilization, outreach, advocacy, and monitoring [36–38]. A report from Kogi State, Nigeria, demonstrated significant success and achievement in using MPPI in an HIV prevention program. According to the study, out-of-school youths comprising male and female were recruited and trained as peer educators who carried out activities which included community dialogue, peer educators recruitment and training, distribution of condoms, and HCT [37]. Similarly, in another report from an intervention among in-school youths in Kwara State, Nigeria, the authors reported the effectiveness of MPPI programming, which addressed behavioral change through the combination of prevention interventions targeted at individuals and communities [36].

We found that communication of HIV prevention messages through parents is an efficient channel for delivering sexuality and HIV messages to their children/wards. Studies have shown that parents/guardians are vital in HIV prevention education of their children/wards [39, 40]. However, cultural barriers often prevent many parents from discussing matters relating to sex with their daughters [41], and many do not have the correct information to pass [42]. They tend to be vague, authoritarian, and indirect about sexual matters [43]. Therefore, programs that target AGYW should look for opportunities to engage parents, especially mothers. Such programs should give parents correct information on HIV and sexuality and help them acquire the necessary communication skills to engage their adolescent children. Previous intervention studies corroborate the findings of our study. A randomized control study in the Bahamas showed improved parent-adolescent communication on sex-related issues, perceived parental monitoring, and the youth's condom use skills and self-efficacy [44]. Further, parental communication interventions have been shown to improve multiple communication domains, including the frequency, quality, intentions, comfort, and self-efficacy for communicating with adolescents [45].

Concerning parent-adolescent SRH communication, interventions that are likely to work are those that specifically train parents in soft communications skills such as talking less, listening more, being less judgmental, and asking more questions in interactions with adolescents about SRH [45]. Efforts should be directed at boosting the self-efficacy of parents (i.e. confidence in the ability to discuss sexual issues with their adolescent) by increasing their knowledge about adolescent SRH [46, 47]. Also, interventions that will increase the motivation of parents to communicate with their adolescents, including both males and females, as well as on all appropriate SRH topics are likely to be effective [48]. An important consideration for program design is to create programs that target parent/caregiver-adolescent pairs as there is

some evidence that they are effective [49]. The interventions that mediate the intergenerational gaps between parents and adolescents by finding a common ground for them are likely to be effective interventions as such gaps have been implicated in poor parent-adolescent communications [50]. Some interventions may be presented via traditional and social media in order to bring them to scale. For example, the use of radio drama has a good history of success in social behavior communication change [51, 52]. Socio-cultural norms impede open sexuality communication, and they are a major impedance to parent-adolescent communication about SRH. Yet, interventions must be sensitive to local cultural and linguistic norms and beliefs for them to gain any traction. Adopting co-creation approaches from the ground up with the people for whom the interventions are targeted, as we did in this study, can help to navigate the norms that are peculiar to different groups.

Peer education is one of the mainstays of behavior communication change in MPPI [35]. Young people are more likely to accept information from their peers based on the peer education approach in HIV prevention strategies [53]. Interventions that address the gap in knowledge and poor perception about HIV need to employ suitable and sufficient channels to deliver messages [54]. We adopted two approaches to engaging peers. One involved AGYW delivering interventions directly to their peers, such as acting as non-traditional outlets for condoms or being mentors and growing their mentoring networks through snowballing. The other involved training selected AGYW to facilitate some programs such as AGYW cell meetings. There were opportunities to acquire income-generating skills and receive HTS and STI counseling as part of the meetings. These approaches are candidates for testing at scale to further demonstrate their effectiveness.

Biomedical approaches usually require trained professionals to conduct pre-and post-test HIV counseling, follow-up counseling and referrals, and STI syndromic management. Adolescents and young people have often faced cultural, structural, personal, or health worker-related barriers to access these services at health facilities on sexual and reproductive health matters [55]. In this study, all our peer-to-peer interventions and youth facilitator-driven interventions that targeted AGYW directly had biomedical components. Here, we used the proverbial one stone to kill two birds underlining the combination prevention approach we adopted, while demonstrating the importance of differential approaches to HIV prevention programming among AGYW. Some other option currently being explored in HTS with some success is testing at non-hospital community-based services such as proprietary and patent medicine vendors (PPMVs) [56]. Also, Nwaozuru et al. [57] showed that adolescents and young people (AYP) have pronounced heterogeneity in HIV testing preferences, including a preference for HIV self-testing.

Structural interventions we adopted included economic empowerment through creating platforms for acquiring income-generating skill sets. Economic empowering of young women can potentially reduce risky sexual behaviors such as transactional and intergenerational sex [58, 59]. We also addressed structural barriers to accessing condoms through the use of non-traditional outlets for condoms. We sought to address gender norms and gender-based violence by engaging local power structures in some locations, such as traditional leaders, fathers, and partners of AGYW, in line with the local context. Our structural approach also included means of bringing HTS and STI services to youth-friendly locations buttressing the point that one size does not fit all. This viewpoint is supported by the fact that HIV prevention programming for AGYW needs to consider the underlying contextual issues that shape risks and vulnerabilities, and must be tackled through a combination of approaches [60]. We also found that HIV testing was the commonest service used following exposure to the various interventions, while family planning was the least. It may be because of the more substantial emphasis on HTS. Further focused research might show if these approaches can be equally valuable for promoting family planning among AGYW.

In all, what is most critical is the combination of intervention approaches. We recommend that interventions brought to scale should have at least a parental communication component, a peer-to-peer component, and innovative ways of removing barriers to accessing condoms. Some of these interventions are easily deployable by local organizations and are not very capital intensive. Challenges may include getting access to testing kits since it is preferable to provide testing services for free. Using empowerment activities that can lead to acquiring skills for income generation is also very important. We used this approach to attract AGYW to meetings where they were offered HIV testing, STI counseling, and referral for STI treatment services. Social media intervention ordinarily would be promising, but such approaches are greatly hampered by internet accessibility and affordability, especially among AGYW of lower socio-economic status. Further studies are needed to test their effectiveness among young people from higher socioeconomic statuses.

This action research presented an excellent opportunity for true collaboration between the community and academia. The research framework ensured that the local priorities and context are taken into account when developing and implementing interventions to reduce the risk of HIV in AGYW. Including the potential beneficiaries of the interventions during intervention model refining processes (planning, implementation, and evaluation cycles) is an improvement over earlier models that were less collaborative in their approach.

This study was not without limitations. The main limitation of this study was the short duration of the action period or intervention. The duration of intervention did not allow for long enough time to observe attributable impact at population levels. However, the interventions developed through a participatory approach with young people and well-tailored to local realities seem to aid the acceptability and accessibility of programs for reducing HIV vulnerability. Also, given that we didn't have a baseline probability of receiving services, nor controlled for confounders or possible contamination effects, we are more restrained about the significant differences found in services used between respondents in the intervention and control LGAs. Finally, we had multiple and varying interventions across the study sites; therefore, we were not able to attribute impact to individual interventions.

## Conclusion

This study used local contextual issues impacting the HIV infection to design customized interventions to reduce HIV vulnerability among AGYW. We demonstrated that young people could participate in developing interventions targeting them. We showed that a combination of behavioral, biomedical, and structural interventions delivered through strategies such as parental communication, peer-to-peer interventions, facilitator-driven interventions, and utilization of non-traditional services for the distribution of condoms have good acceptability among AGYW. These interventions can be location-specific within broader interventions. The location-specific approach allows for more program reach/penetration amongst the target audience. Pursuing approaches developed in participation of adolescents and young people themselves fills the gap of the non-involvement of adolescents and young people in developing, implementing, and evaluating HIV programs targeting them. Future implementation research is required to assess the individual and community contextual factors that can affect the scalability of the interventions to reduce vulnerabilities to HIV infection among AGYW.

## Supporting information

**S1 File. AGYW Action research evaluation questionnaire.**
(DOCX)

**S2 File. AGYW Action research evaluation qualitative guides.**
(DOCX)

**S1 Data. AGYW Action research evaluation quantitative data.**
(DTA)

## Acknowledgments

The authors are thankful to adolescent girls and young women, parents, and other individuals who consented to participate in these studies. We thank the community leaders, authorities, and various health and administrative institutions and officials at the national level, and the various localities we did this study for their permission, support, and guidance to carry out this study.

## Author Contributions

**Conceptualization:** Olujide Arije, Kayode Ijadunola, Olusegun Afolabi, Joshua Aransiola, Godpower Omoregie, Oyebukola Tomori-Adeleye, Oluwole Fajemisin.

**Data curation:** Olujide Arije, Rachel Titus.

**Formal analysis:** Olujide Arije, Joshua Aransiola.

**Funding acquisition:** Kayode Ijadunola.

**Investigation:** Olujide Arije, Ekerette Udoh, Godpower Omoregie, Oyebukola Tomori-Adeleye, Obiarairiuku Ukeme-Edet, Rachel Titus, Adedeji Onayade.

**Methodology:** Olujide Arije, Ekerette Udoh, Olusegun Afolabi, Joshua Aransiola, Oluwole Fajemisin, Adedeji Onayade.

**Project administration:** Olujide Arije, Kayode Ijadunola, Oyebukola Tomori-Adeleye, Obiarairiuku Ukeme-Edet, Rachel Titus, Adedeji Onayade.

**Resources:** Kayode Ijadunola, Godpower Omoregie, Adedeji Onayade.

**Supervision:** Kayode Ijadunola, Olusegun Afolabi, Joshua Aransiola, Oyebukola Tomori-Adeleye, Obiarairiuku Ukeme-Edet, Adedeji Onayade.

**Validation:** Kayode Ijadunola, Godpower Omoregie, Oyebukola Tomori-Adeleye, Obiarairiuku Ukeme-Edet, Oluwole Fajemisin, Rachel Titus.

**Visualization:** Olujide Arije.

**Writing – original draft:** Olujide Arije, Ekerette Udoh, Rachel Titus.

**Writing – review & editing:** Olujide Arije, Ekerette Udoh, Kayode Ijadunola, Joshua Aransiola, Godpower Omoregie, Oyebukola Tomori-Adeleye, Obiarairiuku Ukeme-Edet, Oluwole Fajemisin, Adedeji Onayade.

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
