## [Decision Letter · Decision Letter 0]

11 Aug 2021

PONE-D-21-01166

Minimum prevention package of interventions for reducing vulnerability to HIV among adolescent girls and young women in Nigeria: An action research

PLOS ONE

Dear Dr. Arije,

Thank you for submitting your manuscript to PLOS ONE. After careful consideration, we feel that it has merit but does not fully meet PLOS ONE’s publication criteria as it currently stands. Therefore, we invite you to submit a revised version of the manuscript that addresses the points raised during the review process.

We look forward to receiving your revised manuscript.

Kind regards,

Maria R. Khan, PhD, MPH

Academic Editor

PLOS ONE

Journal Requirements:

2. Please include additional information regarding the survey or questionnaire used in the study and ensure that you have provided sufficient details that others could replicate the analyses. For instance, if you developed a questionnaire as part of this study and it is not under a copyright more restrictive than CC-BY, please include a copy, in both the original language and English, as Supporting Information.  If the original language is written in non-Latin characters, for example Amharic, Chinese, or Korean, please use a file format that ensures these characters are visible.

3. Please state whether you validated the questionnaire prior to testing on study participants. Please provide details regarding the validation group within the methods section.

4. Please include a copy of the interview guide used in the study, in both the original language and English, as Supporting Information, or include a citation if it has been published previously.

5. You indicated that you had ethical approval for your study. In your Methods section, please ensure you have also stated whether you obtained consent from parents or guardians of the minors included in the study or whether the research ethics committee or IRB specifically waived the need for their consent.

6. We note that the grant information you provided in the ‘Funding Information’ and ‘Financial Disclosure’ sections do not match. 

7. In your Data Availability statement, you have not specified where the minimal data set underlying the results described in your manuscript can be found. PLOS defines a study's minimal data set as the underlying data used to reach the conclusions drawn in the manuscript and any additional data required to replicate the reported study findings in their entirety. All PLOS journals require that the minimal data set be made fully available. For more information about our data policy, please see http://journals.plos.org/plosone/s/data-availability.

Reviewers' comments:

Reviewer's Responses to Questions

**Comments to the Author**

1. Is the manuscript technically sound, and do the data support the conclusions?

Reviewer #1: Partly

Reviewer #2: Yes

2. Has the statistical analysis been performed appropriately and rigorously? 

Reviewer #1: Yes

Reviewer #2: Yes

3. Have the authors made all data underlying the findings in their manuscript fully available?

Reviewer #1: No

Reviewer #2: Yes

4. Is the manuscript presented in an intelligible fashion and written in standard English?

Reviewer #1: Yes

Reviewer #2: Yes

5. Review Comments to the Author

Reviewer #1: Review

Abstract

The background section in the abstract reads as a method section. The background section should reflect studies that have examined similar interventions and the authors should discuss the gap in the literature and how a study like this fills that gap.

Introduction

The authors state that “Despite the myriad programs and agencies offering HIV-related interventions in the country”- The authors should mention 2-3 of these programs and describe the HIV-related interventions these programs offer in comparison to what is being proposed in this study.

The authors should either cite similar studies that have used action research methodology for HIV related research (in a different population, age group, or another disease) and mention how this study differs OR the authors should state that this will be the first study to their knowledge that uses the action methodology to address this burden.

Line 77 needs a citation – “The Breakthrough Series (BTS) collaborative lends itself to the ideals of action research.”

The authors define BTS well and state that the BTS approach has been used extensively but the authors fail to provide examples of studies that have used them. The authors can provide 1 or 2 examples in which the BTS method has been used

The last paragraph of this section should have the objective of the paper and the hypothesis. The authors should consider a revision.

Methods

The authors should clearly state the study design e.g. this is a qualitative study using xx grounded approach/methodology.

The authors state that “We purposively selected two local government areas (LGA) as implementation sites and one as the control site in each study location” but the authors fail to give a reason why?- The authors should consider providing a reason for this.

The authors mention “local BTS teams planned the specific interventions to address the change topics identified in each study state” But the authors do not report on who makes up these local BTS teams. The description of the teams needs to be reported first before the activities of these teams are described. The authors should consider describing these BTS teams and the type of stakeholders that constitute each of these teams.

A dedicated section for recruitment and population description needs to be included in the methods section i.e. how were youth participants and other local stakeholders recruited into the study. (please explain the process of recruitment). Describe all participants and stakeholders who participated in the study.

What was the duration of each action period (3 months? 4 months?). The actual duration needs to be stated as well as the timing (2019-2020?)

Line 59-60 needs to be revised “There were two action periods in this study, lasting three and two each.”- Do the authors mean to write 2- 3 months? This needs to be clarified

The section labeled household survey is reporting on demographics, which should be in the results section. The household survey should simply describe the methodology that was employed and should not report on results (including demographics) of the survey. E.g. line 69 “The study population was females of the age range of 15 to 24 years old” These errors are also seen in lines 193-195. All these should be in the results section, not the methods section.

The authors need to define acceptability (what do the authors hypothesize acceptability for this study will look like). If possible the authors should also cite their definition from the literature

The authors need to define accessibility (what do the authors hypothesize accessibility for this study will look like). If possible the authors should also cite their definition from the literature

Results

For each theme/domain idea, the authors should consider including quotes from participants in describing some of the results. This is only evident in the parental communication intervention themes onwards, previous themes like Peer-to-peer communication interventions and Youth facilitator-driven interventions do not have quotes. The authors should consider revising this to include quotes from stakeholders.

Discussion

The authors state that “Several authors have successfully implemented HIV prevention interventions based on the MPPI strategy among in-school and out-of-school adolescents, including community mobilization, outreach, advocacy, and monitoring”- But the authors do not describe some of these studies. The authors should consider describing 1 or 2 of these studies.

The authors should report on or cite other studies/programs from the literature (in other populations, countries, etc.) that have targeted AGYW and their parents for matters relating to sex or HIV prevention, etc. OR the authors must state that no such study exists in the literature and this study will be the first of its kind.

Line 445-446 needs a citation from the literature- “These buttress the point that one size will not fit all, and context is (nearly) everything in HIV prevention programming for AGYW.”

The authors must include a dedicated section in the discussion section to describe and discuss the study limitations. This should not be in the conclusion.

Conclusion

The conclusion section should simply summarize the overall findings and what future research should explore, the study limitations should be removed from this section and have a dedicated section in the discussion section.

Overall Summary and reviewer final comments

Overall this is a good paper, which takes an innovative approach to the topic. The authors should consider the revision suggestions above and also consider a clean copy edit of the manuscript as there are a few, run-on sentences, tense confusions, and grammatical errors which need to be fixed using spell check or software like Grammarly. Once these edits are incorporated this paper should be accepted for publication.

Reviewer #2: This manuscript provides an excellent example of a true collaboration between the community and academia. The authors utilize a framework to ensure that the local priorities and context are taken into account when developing and implementing an intervention to reduce the risk of HIV in AGYW. Refining models of the intervention planning, implementation, and evaluation cycle to include those the intervention is meant to serve is laudable. However, my appreciation of the contribution of this research might be enhanced by a clearer presentation of methods, a more rigorous analysis of data (or at least a clear acknowledgment of its limitation), and a possible reframing of the discussion.

I found it difficult to follow the connection between action research, breakthrough series, a learning collaborative approach, and a plan-do-study-act cycle. I also found it challenging to understand the relationship between change packages, the combination prevention intervention approach, intervention models, change ideas, and minimum prevention package of interventions. Additionally, I believe it would be helpful to have additional information about what measures were evaluated when. For example, what was assessed during the "monitoring and evaluation component"? What did local CBOs evaluate to test change ideas? What criteria were used to determine "whether to adapt, adopt, abandon, increase scale, or test under different conditions?" What was measured at baseline and follow-up, and what was considered an "outcome of the specific interventions" and evaluated only at follow-up?

In terms of analysis, given that we don't know the baseline probability of receiving services, and the authors did not control for confounders or possible contamination effects, I'm wary of ascribing too much salience to the significant differences found between services used between the intervention LGAs and controls. Also, as multiple interventions were available in the LGAs, I believe the text would be improved by describing how authors ascertained that services used were due to a particular intervention. Finally, in the discussion, the authors state this their approach "helped improve the acceptability and accessibility of programs." I would be careful to overstate this point. While they've shown this approach results in acceptable and accessible programs, without comparing it to other models of intervention development and implementation, it's impossible to say whether it was an "improvement." However, I think the authors miss an opportunity to highlight how this collaborative approach, which engages the community and takes context into account, might be an improvement over other models. Consider framing the discussion to illuminate the relative benefits of this approach.

6. PLOS authors have the option to publish the peer review history of their article (what does this mean?). If published, this will include your full peer review and any attached files.

Reviewer #1: No

Reviewer #2: No

---

## [Author Response · Author response to Decision Letter 0]

30 Sep 2021

Reviewer #1

Abstract

The background section in the abstract reads as a method section. The background section should reflect studies that have examined similar interventions and the authors should discuss the gap in the literature and how a study like this fills that gap.

***The abstract has been revised

Introduction

The authors state that “Despite the myriad programs and agencies offering HIV-related interventions in the country”- The authors should mention 2-3 of these programs and describe the HIV-related interventions these programs offer in comparison to what is being proposed in this study.

***Some HIV-related interventions in the country have been mentioned

The authors should either cite similar studies that have used action research methodology for HIV related research (in a different population, age group, or another disease) and mention how this study differs OR the authors should state that this will be the first study to their knowledge that uses the action methodology to address this burden.

Some similar studies have been cited

Line 77 needs a citation – “The Breakthrough Series (BTS) collaborative lends itself to the ideals of action research.”

Citation has been added

The authors define BTS well and state that the BTS approach has been used extensively but the authors fail to provide examples of studies that have used them. The authors can provide 1 or 2 examples in which the BTS method has been used.

***Some examples have been provided

The last paragraph of this section should have the objective of the paper and the hypothesis. The authors should consider a revision.

***Objective of study has been added to the last paragraph

Methods

The authors should clearly state the study design e.g. this is a qualitative study using xx grounded approach/methodology.

***The study design has been stated

The authors state that “We purposively selected two local government areas (LGA) as implementation sites and one as the control site in each study location” but the authors fail to give a reason why?- The authors should consider providing a reason for this.

***Bases of selection of study locations have been stated

The authors mention “local BTS teams planned the specific interventions to address the change topics identified in each study state” But the authors do not report on who makes up these local BTS teams. The description of the teams needs to be reported first before the activities of these teams are described. The authors should consider describing these BTS teams and the type of stakeholders that constitute each of these teams.

***Members of the BTS teams have been included.

A dedicated section for recruitment and population description needs to be included in the methods section i.e. how were youth participants and other local stakeholders recruited into the study. (please explain the process of recruitment). Describe all participants and stakeholders who participated in the study.

***A dedicated section for recruitment section has been added

What was the duration of each action period (3 months? 4 months?). The actual duration needs to be stated as well as the timing (2019-2020?)

Duration information has been included

Line 59-60 needs to be revised “There were two action periods in this study, lasting three and two each.”- Do the authors mean to write 2- 3 months? This needs to be clarified

***Clarification has been provided 

The section labeled household survey is reporting on demographics, which should be in the results section. The household survey should simply describe the methodology that was employed and should not report on results (including demographics) of the survey. E.g. line 69 “The study population was females of the age range of 15 to 24 years old” These errors are also seen in lines 193-195. All these should be in the results section, not the methods section.

***These errors have been corrected

The authors need to define acceptability (what do the authors hypothesize acceptability for this study will look like). If possible the authors should also cite their definition from the literature.

***Acceptability has been defined

The authors need to define accessibility (what do the authors hypothesize accessibility for this study will look like). If possible the authors should also cite their definition from the literature

***Accessibility has been defined

Results

For each theme/domain idea, the authors should consider including quotes from participants in describing some of the results. This is only evident in the parental communication intervention themes onwards, previous themes like Peer-to-peer communication interventions and Youth facilitator-driven interventions do not have quotes. The authors should consider revising this to include quotes from stakeholders.

***The result section is divided into 2 part: ‘The Change ideas’ and ‘Reach and uptake of the change ideas’. The first part was a brief narration of the various interventions (change ideas) that were developed and implemented across the different study locations. The second aspect described report findings from data collected form study participants. Kindly observe that all of the theme/domain ideas have quotes from participants in describing some of the results

Discussion

The authors state that “Several authors have successfully implemented HIV prevention interventions based on the MPPI strategy among in-school and out-of-school adolescents, including community mobilization, outreach, advocacy, and monitoring”- But the authors do not describe some of these studies. The authors should consider describing 1 or 2 of these studies.

***Additional studies have been described

The authors should report on or cite other studies/programs from the literature (in other populations, countries, etc.) that have targeted AGYW and their parents for matters relating to sex or HIV prevention, etc. OR the authors must state that no such study exists in the literature and this study will be the first of its kind.

Additional studies/programs from the literature have been cited

Line 445-446 needs a citation from the literature- “These buttress the point that one size will not fit all, and context is (nearly) everything in HIV prevention programming for AGYW.”

***A citation from the literature has been added

The authors must include a dedicated section in the discussion section to describe and discuss the study limitations. This should not be in the conclusion.

***The study limitations have been included in the discussion section

Conclusion

The conclusion section should simply summarize the overall findings and what future research should explore, the study limitations should be removed from this section and have a dedicated section in the discussion section.

***The conclusion has been modified accordingly

Reviewer #2: Review

However, my appreciation of the contribution of this research might be enhanced by a clearer presentation of methods, a more rigorous analysis of data (or at least a clear acknowledgment of its limitation), and a possible reframing of the discussion.

I found it difficult to follow the connection between action research, breakthrough series, a learning collaborative approach, and a plan-do-study-act cycle. 

I also found it challenging to understand the relationship between change packages, the combination prevention intervention approach, intervention models, change ideas, and minimum prevention package of interventions. 

***The connection among the above concepts have been clarified especially under the ‘Implementation approach’, 

'Learning Sessions' and 'Action Session'

Additionally, I believe it would be helpful to have additional information about what measures were evaluated when. For example, 

• what was assessed during the "monitoring and evaluation component"? 

***This information has been included

• What did local CBOs evaluate to test change ideas?

***The word test has been replaced with implement which we used interchangeably, and the role of the CBOs further clarified 

• What criteria were used to determine "whether to adapt, adopt, abandon, increase scale, or test under different conditions?" 

***This information has been included

• What was measured at baseline and follow-up, and what was considered an "outcome of the specific interventions" and evaluated only at follow-up?

***This information has been included

In terms of analysis, given that we don't know the baseline probability of receiving services, and the authors did not control for confounders or possible contamination effects, I'm wary of ascribing too much salience to the significant differences found between services used between the intervention LGAs and controls. 

***This detail has been included in the text as a limitation

Also, as multiple interventions were available in the LGAs, I believe the text would be improved by describing how authors ascertained that services used were due to a particular intervention. 

***The research assistants were trained to describe the intervention implemented in the different study locations to reduced misclassification bias. This information has been added 

Finally, in the discussion, the authors state this their approach "helped improve the acceptability and accessibility of programs." I would be careful to overstate this point. While they've shown this approach results in acceptable and accessible programs, without comparing it to other models of intervention development and implementation, it's impossible to say whether it was an "improvement." 

***We have modified our statement to be more conservative.

However, I think the authors miss an opportunity to highlight how this collaborative approach, which engages the community and takes context into account, might be an improvement over other models. Consider framing the discussion to illuminate the relative benefits of this approach. 

***We have attempted to do highlight the collaborative approach

---

## [Decision Letter · Decision Letter 1]

18 Apr 2022

PONE-D-21-01166R1Minimum prevention package of interventions for reducing vulnerability to HIV among adolescent girls and young women in Nigeria: An action researchPLOS ONE

Dear Dr. Arije,

Thank you for submitting your manuscript to PLOS ONE. After careful consideration, we feel that it has merit but does not fully meet PLOS ONE’s publication criteria as it currently stands. Therefore, we invite you to submit a revised version of the manuscript that addresses the points raised during the review process.

We look forward to receiving your revised manuscript.

Kind regards,

Tai-Heng Chen, M.D.

Academic Editor

PLOS ONE

Journal Requirements:

Reviewers' comments:

Reviewer's Responses to Questions

**Comments to the Author**

1. If the authors have adequately addressed your comments raised in a previous round of review and you feel that this manuscript is now acceptable for publication, you may indicate that here to bypass the “Comments to the Author” section, enter your conflict of interest statement in the “Confidential to Editor” section, and submit your "Accept" recommendation.

Reviewer #3: (No Response)

Reviewer #4: (No Response)

Reviewer #5: (No Response)

2. Is the manuscript technically sound, and do the data support the conclusions?

Reviewer #3: Yes

Reviewer #4: Partly

Reviewer #5: Yes

3. Has the statistical analysis been performed appropriately and rigorously? 

Reviewer #3: N/A

Reviewer #4: I Don't Know

Reviewer #5: Yes

4. Have the authors made all data underlying the findings in their manuscript fully available?

Reviewer #3: Yes

Reviewer #4: Yes

Reviewer #5: Yes

5. Is the manuscript presented in an intelligible fashion and written in standard English?

Reviewer #3: Yes

Reviewer #4: No

Reviewer #5: No

6. Review Comments to the Author

Reviewer #3: This paper provides important evidence and discussion of ways in which to develop combination interventions that are context appropriate, and acceptable to their target populations and the communities they are embedded in – with the buy in and participation of the target population and broader community. However, I am not sure the authors have sufficiently discussed the potentially problematic contextual elements, community dynamics or beliefs, that might hamper AGYW sexual health, sexual rights, sexual autonomy and empowerment – such as entrenched gender inequities, “traditional beliefs”, or sexual communication norms that inhibit/prohibit open communication and access to information.

Whilst the authors state that programmes should “engage parents” and provide parents with “correct information on HIV and sexuality and help them acquire the necessary communication skills to engage their adolescent children” – I don’t see discussions of potential sociocultural barriers towards doing this - what kinds of challenges or resistance there may be at the community level, and how interventions might circumvent/address these.

General comments

- I don’t see discussion of or reflection on diversity in sexuality. Was there reflection on heteronormativity in socio-cultural contexts, and how this was embedded in parent-adolescent sexuality communication?

- How does the centrality and positioning of the church as a mechanism through which sexuality communication and education occurs shape the content and framing of sex? E.g. discussions of sexual pleasure and empowerment versus ‘abstinence until marriage’ etc? It would be good to see some reflection on this.

- Norms around sexuality communication are mentioned – e.g. appropriateness of discussing sexual organs/genitalia, and euphemistic nature of language prohibiting open and explicit discussion – how does this impact young people’s knowledge and sexual decision making?

- There is evidence of parental resistance to comprehensive sexuality education in this context (CSE)? Was this also the case at a broader community and institutional level (e.g schools, churches)? How does this impact intervention delivery?

Specific comments:

Page 2

- Line 19: re-order sentence – “AGYW in Nigeria…”

Page 5

- Line 100: why is “problem’s” possessive? Incorrect use of apostrophe

Page 7

- Lines 134-137: check brackets are closed

Page 17

- Line 370: surely a 21 year old female is a “young woman”, not a “girl”? Refer to standard definitions of “girls” vs “women”, and age brackets of AGYW

Reviewer #4: The manuscript is written in standard English, but I struggled with some ambiguous sentences and flow.

Reviewer #5: Dear Editor,

Thank you so much for giving me an opportunity to review the manuscript. First, I had to read comments of earlier reviewers. The authors have addressed most concerns raised by earlier reviewers. I make the following comments:

Title: The authors might consider revising the title to include or replace ‘minimum’ with ‘combined’ to tie with the objective of the study. In addition, the authors might consider including adolescent boys and young men in the title because results show that these people participated in the study too.

Abstract: Lines 22-25 read like methods section. Authors should consider including recommendation(s) [it can be within the conclusion]. Proofread for typo errors e.g. Line 19 ‘adolescents’ ‘[adolescent]’.

Introduction: First statement needs revision – whether it is about adolescent girls or adolescents in general. The section needs proofreading for grammatical errors. Transitions are lacking between some paragraphs.

Methods and Materials: Lines 217-218 need citation(s).

Results: Presentation of results is not clear especially at the beginning. I suggest focusing on the results and removing some background information. Some information might be used to strengthen the Methods and Materials section.

Proofread for typo, grammar and spelling errors e.g. Line 423, line 458, line 497, line 533, line 545 etc.

It is only when I get to line 479 that I begin reading of young men. Again, these are not reflected in the title, abstract, introduction, methods and materials sections. Provide an example to lines 484-487 or cite the literature. Some quotes are not identified (have no persons who said them). Other quotes can be shortened.

Discussion and Conclusion sections are clear to me.

Reference section needs editing for consistency including for dates.

Others: Check whether it is HIV/AIDS or [HIV and AIDS]

Thank you.

7. PLOS authors have the option to publish the peer review history of their article (what does this mean?). If published, this will include your full peer review and any attached files.

Reviewer #3: No

Reviewer #4: No

Reviewer #5: **Yes: **Wilfred Masebo

---

## [Author Response · Author response to Decision Letter 1]

9 Sep 2022

Reviewer #3: 

General comments

I don’t see discussion of or reflection on diversity in sexuality. Was there reflection on heteronormativity in socio-cultural contexts, and how this was embedded in parent-adolescent sexuality communication?

- How does the centrality and positioning of the church as a mechanism through which sexuality communication and education occurs shape the content and framing of sex? E.g. discussions of sexual pleasure and empowerment versus ‘abstinence until marriage’ etc? It would be good to see some reflection on this.

- Norms around sexuality communication are mentioned – e.g. appropriateness of discussing sexual organs/genitalia, and euphemistic nature of language prohibiting open and explicit discussion – how does this impact young people’s knowledge and sexual decision making?

- There is evidence of parental resistance to comprehensive sexuality education in this context (CSE)? Was this also the case at a broader community and institutional level (e.g schools, churches)? How does this impact intervention delivery?

Response: All interventions in our study were developed, tested and refined iteratively through co-creation with AGYW as well as other stakeholders in each local study site. These interventions were contextual and we didn’t find substantial evidence of parental resistance to engaging their children/wards on sexual education particularly in the intervention locations, or of AGYW resistance in receiving sexual education from their parents as documented in our findings. Moreover we were careful to allow the community members have substantial inputs in order not to impose the world view of the researchers on them. What we have reported is the findings using the action research approach findings from assessing exposure to the interventions implemented. These interventions were done in very limited scale but future studies can help to explore broader context and implication to parental engagement in the sexuality of the adolescents and young people. It is important for us as researchers to continue to be culturally sensitive, and not to foist what might be interpreted as western ideas on non-western local communities given that different societies are at different levels or phases of social progressiveness.

Specific comments:

Page 2

- Line 19: re-order sentence – “AGYW in Nigeria…”

Response: This has been corrected

Page 5

- Line 100: why is “problem’s” possessive? Incorrect use of apostrophe

Response: This has been corrected

Page 7

- Lines 134-137: check brackets are closed

Response: This has been corrected

Page 17

- Line 370: surely a 21 year old female is a “young woman”, not a “girl”? Refer to standard definitions of “girls” vs “women”, and age brackets of AGYW

Response: This has been corrected

Reviewer #4

Thanks for your work on updating this paper and the presentation of an important, interesting study. I believe substantial work is needed to the methods sections (as well as other sections). There are a LOT of components of this study and it is going to be critical to make sure to be super specific in outlining each component and detailing how they all fit together. 

• Implementation of these change ideas is a central component and I believe this paper could vastly benefit from incorporation of some implementation science framework and/or theory. 

Response: We consider this suggestion important but we are weary of adopting an implementation research framework since that was not how the intervention was carried out. We prefer to stay fully within an action research context which we consider as the proper way to view and report the research.

• Would also recommend a conceptual framework to outline how all the pieces fit together, the mechanism at play, and the implementation outcomes assessed.

Response: We have included a conceptual framework

• Highlighting how this work is embedded within the larger HIV response within Nigeria is also super important. 

Response: We have included a statement to highlight this

• The abstract highlights that vulnerability and socio-cultural contexts vary across Nigeria. 

Would expand upon this in the introduction and highlight some of these varying differences across the nation (also providing rationale to the selection of study locations). 

Response: We provided rationale for selection of the study locations under ‘Study setting’. 

• The abstract notes that this study developed a minimum HIV prevention package for AGYW. How did this differ from package of care put forth within the Nigerian NPP 2010-2012 and the MPPI? Does the country currently have guidelines or recommendations for AYGW? How are these guidelines and recommendations similar and/or different from the FLHE curriculum or national Youth service corps program? These components feel disaggregated as they’re quite separated from one another within the introduction. Would recommend reframing. 

Response: The country is still in the process of developing a scalable guide for the implementation of community-based HIV programs focused on AYP in Nigeria (personal communication). The package of care put forth in the Nigerian NPP 2010-2012 and the MPPI are generic while the prevention package we present in this study are adaptive and contextualized. The FLHE curriculum and National Youth Service Corps peer-educator program are earlier approaches to scale-up HIV prevention and are not necessarily combination prevention approaches.

• The aim of the study put forth in the abstract differs slightly (from my understanding) from the aim of the study put forth in the last sentences of the introduction. Would recommend harmonizing. 

Response: We have revised this

• It is helpful to see the citation of similar studies that have leveraged action research methods, per reviewer #1’s suggestion. Would recommend going a step further to outline the key lessons learned or implications for this work, ie connect it back to the ongoing study to see seems a bit more integrated and appropriate within the introduction. 

Response: We have further revised this section

• In the methods it states “This research has a mixed-method design involving a cross-sectional descriptive study, focus group discussions with AGYW, and one-on-one interviews with selected key informants.” I find this a tad confusing. Specifically, can you specify which methods are quantitative verse qualitative? Are the one-on-one interviews, in-depth qualitative interviews?

Response: This has been revised

• Great to see you’ve incorporated justification for study site selection. Would be helpful to outline in the introduction the differences and distribution of HIV prevalence among AGYW across states of Nigeria. 

Response: We have updated the information in the “Study setting’ section

• In lines 153-156 it states that interventions were selected based on estimated youth population and the absence of existing interventions. Were there prior studies conducted to provide size estimations of AGYW in these areas? Please specify. Also, what was done previously to document and quantify existing interventions within these areas? Please specify. How recently were these things done?

Response: This has been revised

• You define a comprehensive change package. But how do you define “desired change.” Would recommend looking at specific implementation outcomes to quantify. 

Response: Desired change was contextual for each ‘change idea’ implemented and was evaluated using the decision rule for implemented change ideas (Table 3 in the manuscript)

• Impressive how collaborative teams were across levels and programs. Would you add a few sentences to outline how communication occurred across team and how consistency in implementation was guaranteed? 

Response: We have introduced a chart to show the line of communication

• Who conducted the scoring of the change topics? How was this process conducted and how was the prioritization matrix utilized? Was there 1 matrix overall or per study site? Did this happen once or iteratively? Were AGYW involved in the process?

Response: Scoring of the change topics using the prioritization matrix was conducted during the learning sessions which held separately in each study location when the local BTS needed to decide on which change topic to intervene for. AGYW were members of the local BTS team (stated earlier under section titled ‘BTS Collaborative management’.

• It’s noted that the third learning session was to review the implementation process. How was this done? How was implementation planned and monitored?

Response: We used the monitoring and evaluation reports from each of the interventions implemented. We have revised this section in the manuscript

• You outline that location CBOs were engaged in the implementation of the change ideas. How were CBOs trained? How was uniform implementation ensured? How was fidelity of the implementation assessed? Did all CBOs have similar staffing and resources?

Response: Because of the diversity of interventions across the different study locations, the CBOs engaged did not have similar staffing and resources. Even then, it was not necessary to have uniformity across the study locations since different interventions were implemented in different places. CBOs were engaged according to the need in each study site and the research provided site-specific training and support as was necessary for the CBOs that were engaged

• Can you specify what you mean by a “A multi-stage sampling technique was employed to sample participants at enumeration area-level and at household level.” – I am struggling to understand. Was the sampling the same for baseline and endline?

Response: The sampling method was the same at baseline and at endline but it was not a panel data. Different set of respondents were recruited from the same locations at the different times although a few participated in both the baseline and endline survey (but not be design). We have refrained from dwelling on the methodology of the baseline assessment since we did not report it in this current report.

• What does “Although the overall evaluation of the action research involved a baseline and end-line assessment, the outcome of specific interventions was accessed only at the end-line since those interventions were not already designed at the start of the research and would not fit a pre-post assessment” mean? How was engagement in HIV prevention services assess at the baseline survey?

Response: The action research design we adopted does not fit a pre-post assessment because the specific interventions implemented were designed after there was a baseline assessment. Findings from the baseline assessment were part of the sources of information for designing the various intervention. Because these interventions were non-exist as at baseline, no specifc data can be collected about them at baseline. In this report, we did not include any baseline data. We have revised the section to more properly reflect what we did.

• Were respondents reimbursed for participation in the survey? Were participants reimbursed for qualitative interviews?

Response: All FGD and IDI participants received transportation reimbursement of ₦1000 each ($1≈₦360, 2017) Key informants were not reimbursed because interviews took place in their various offices. 

• What was the sample size for the endline survey?

Response: The total sample size was 4308, and we have updated the manuscript in this respect

• Were qualitative interviews conducted with AGYW who participated in the interventions specifically? All interventions or some? How was sampling done for the qualitative interviews? What was the sample size? There is mention of snowball sampling within the results but it is unclear for which component and this level of detail should be included within the methods. 

Response: We have updated this section to include the various categories of participants in the FGD. We included both participants and non-participants of the action research implementation component in the FGD. Snowballing was used as part of the intervention component not evaluation component of the action research. We have moved the section describing the ‘Change ideas’ to the methodology.

• How was intervention uptake quantified? How was the denominator determined?

Response: The intervention uptake was quantified as a proportion of participants that reported exposure to the various groups of interventions. The denominator in each study state was the total number of respondents in each state

• Were all qualitative interviews coded with the same codebook? How was coding discrepancies resolved among the 4 coders? Were interview singly or doubly coded? 

Response: A generic codebook was used across all transcripts but adapted to suit different groups of study participants. We have also updated this session

• Was there any additional eligibility criteria for participants? What about if AGYW were living with HIV, were they included? What was the eligibility criteria for key informants?

Response: We have revised this section

Reviewer #5

Thank you so much for giving me an opportunity to review the manuscript. First, I had to read comments of earlier reviewers. The authors have addressed most concerns raised by earlier reviewers. I make the following comments:

Title: The authors might consider revising the title to include or replace ‘minimum’ with ‘combined’ to tie with the objective of the study. In addition, the authors might consider including adolescent boys and young men in the title because results show that these people participated in the study too.

Response: We agree with your suggestion to replace minimum with combination. However, our interventions targeted AGYW, and boys and young men were only included in the study as stakeholders in the vulnerability of AGYW to HIV, Hence, we consider it more appropriate to retain the AGYW focus

Abstract: Lines 22-25 read like methods section. Authors should consider including recommendation(s) [it can be within the conclusion]. Proofread for typo errors e.g. Line 19 ‘adolescents’ ‘[adolescent]’.

Response: We have revised this

Introduction: First statement needs revision – whether it is about adolescent girls or adolescents in general. The section needs proofreading for grammatical errors. Transitions are lacking between some paragraphs. 

Response: 

We have revised this

Methods and Materials: Lines 217-218 need citation(s).

Response: We have revised this

Results: Presentation of results is not clear especially at the beginning. I suggest focusing on the results and removing some background information. Some information might be used to strengthen the Methods and Materials section. 

Response: We have moved the section describing the change ideas to the methodology

Proofread for typo, grammar and spelling errors e.g. Line 423, line 458, line 497, line 533, line 545 etc. 

Response: We have revised this

It is only when I get to line 479 that I begin reading of young men. Again, these are not reflected in the title, abstract, introduction, methods and materials sections. Provide an example to lines 484-487 or cite the literature. Some quotes are not identified (have no persons who said them). Other quotes can be shortened.

Response: The interventions we have described in our manuscript are focused on AGYW. Young men were included in the action research only as stakeholders like other stakeholders such as the gatekeepers (religious and community leaders) and key informants (health workers, program officers etc).

Discussion and Conclusion sections are clear to me.

Reference section needs editing for consistency including for dates. 

Response: We have revised the reference section

Others: Check whether it is HIV/AIDS or [HIV and AIDS]

Response: We have updated this to HIV and AIDS

Thank you.

---

## [Decision Letter · Decision Letter 2]

7 Oct 2022

PONE-D-21-01166R2Combination prevention package of interventions for reducing vulnerability to HIV among adolescent girls and young women in Nigeria: An action researchPLOS ONE

Dear Dr. Arije,

Thank you for submitting your manuscript to PLOS ONE. After careful consideration, we feel that it has merit but does not fully meet PLOS ONE’s publication criteria as it currently stands. Therefore, we invite you to submit a revised version of the manuscript that addresses the points raised during the review process.

We look forward to receiving your revised manuscript.

Kind regards,

Tai-Heng Chen, M.D.

Academic Editor

PLOS ONE

Journal Requirements:

Reviewers' comments:

Reviewer's Responses to Questions

**Comments to the Author**

1. If the authors have adequately addressed your comments raised in a previous round of review and you feel that this manuscript is now acceptable for publication, you may indicate that here to bypass the “Comments to the Author” section, enter your conflict of interest statement in the “Confidential to Editor” section, and submit your "Accept" recommendation.

Reviewer #3: (No Response)

2. Is the manuscript technically sound, and do the data support the conclusions?

Reviewer #3: Yes

3. Has the statistical analysis been performed appropriately and rigorously? 

Reviewer #3: N/A

4. Have the authors made all data underlying the findings in their manuscript fully available?

Reviewer #3: Yes

5. Is the manuscript presented in an intelligible fashion and written in standard English?

Reviewer #3: Yes

6. Review Comments to the Author

Reviewer #3: This manuscript has been much improved. I believe it would be ready for publication once a few last changes have been made as suggested below:

1. Line 163: “Combination HIV prevention is the recommended approach for comprehensive HIV prevention.” – does this sentence have value? Would benefit from being rephrased.

2. Line 192: conceptual framework “used by / proposed by” Chimbindi et al ?

3. Can the authors make more specific and actionable recommendations based on their findings? For example strategies for supporting parent-adolescent SRH communication.

4. How much do parents’ own communication skills and SRH knowledge play a role? How can interventions bolster parents’ communication skills – such as those in the Akers paper the author cites.

Also in: DiIorio et. (2000). Social cognitive factors associated with mother-adolescent communication about sex. Journal of Health Communication, 5(1), 41–51. doi:10.1080/108107300126740

5. How does parents’ own self-efficacy play a role? How can interventions best provide parents/caregivers with a knowledge base and skill-set, to enhance their motivation and confidence in communicating around SRH topics?

See discussion on this in:

- Guilamo-Ramos et al. (2008). Parent-adolescent communication about sexual inter- course: An analysis of maternal reluctance to communicate. Health Psychology, 27(6), 760–769. doi:10.1037/a0013833

- Seif et al. (2018). Caretaker-adolescent communication on sexual and reproductive health: A cross- sectional study in Unguja-Tanzania Zanzibar. BMC Public Health, 18(31), 1–13. doi:10.1186/ s12889-017-4591-2

6. How can interventions empower parent/caregivers in order to increase their ability to be responsive to the needs of adolescents and support them to make safe and informed decisions about sex? Can the authors make recommendations pertaining to this issue.

See discussion on this in:

- Grossman et al. (2017). “We talked about sex.” “No, we didn’t”: Exploring adolescent and parent agreement about sexuality communication. American Journal of Sexuality Education. doi:10. 1080/15546128.2017.1372829

- Nilsson et al. (2020). Obstacles to intergenerational communication in care- givers’ narratives regarding young people’s sexual and reproductive health and lifestyle in rural South Africa. BMC Public Health, 20(791). doi:10.1186/s12889-020-08780-9

7. If socio-cultural norms impede open sexuality communication, is there a strategy for helping parents/elders to understand the implications of their attitudes and communication style on adolescent sexual decision making? Specific recommendations relating to this issue would be helpful.

See discussion of this in:

- AVAC. (2018). Breaking the cycle of transmission: Increasing adoption of and adherence to effective HIV prevention among high-risk adolescent girls and young women.

- Duby et al. (2022) ‘I can't go to her when I have a problem’: sexuality communication between South African adolescent girls and young women and their mothers, SAHARA-J: Journal of Social Aspects of HIV/AIDS, 19:1, 8-21, https://doi.org/10.1080/17290376.2022.2060295

8. How can policy makers and programme designers ensure that interventions are designed and implemented in ways that are sensitive to local cultural and linguistic norms and beliefs? See the Nilson and Duby references cited above for discussion of this. Including recommendations to consider this would be beneficial.

7. PLOS authors have the option to publish the peer review history of their article (what does this mean?). If published, this will include your full peer review and any attached files.

Reviewer #3: No

---

## [Author Response · Author response to Decision Letter 2]

12 Nov 2022

Dear Sir/Ma

Below is a point by point responses to the comments provided to us.

We look forward to finalizing the manuscript.

Thanks once again.

OAA

Response to reviewers

1. Line 163: “Combination HIV prevention is the recommended approach for comprehensive HIV prevention.” – does this sentence have value? Would benefit from being rephrased.

‘Combination HIV prevention’ is a specific terminology in HIV prevention and care. However we have revised this line to provide more clarity without loss of meaning.

2. Line 192: conceptual framework “used by / proposed by” Chimbindi et al? 

This line has been corrected

3. Can the authors make more specific and actionable recommendations based on their findings? For example, strategies for supporting parent-adolescent SRH communication.

4. How much do parents’ own communication skills and SRH knowledge play a role? How can interventions bolster parents’ communication skills – such as those in the Akers paper the author cites. Also in: DiIorio et. (2000). Social cognitive factors associated with mother-adolescent communication about sex. Journal of Health Communication, 5(1), 41–51. doi:10.1080/108107300126740

5. How does parents’ own self-efficacy play a role? How can interventions best provide parents/caregivers with a knowledge base and skill-set, to enhance their motivation and confidence in communicating around SRH topics?

See discussion on this in:

a. - Guilamo-Ramos et al. (2008). Parent-adolescent communication about sexual inter- course: An analysis of maternal reluctance to communicate. Health Psychology, 27(6), 760–769. doi:10.1037/a0013833

b. - Seif et al. (2018). Caretaker-adolescent communication on sexual and reproductive health: A cross- sectional study in Unguja-Tanzania Zanzibar. BMC Public Health, 18(31), 1–13. doi:10.1186/ s12889-017-4591-2

6. How can interventions empower parent/caregivers in order to increase their ability to be responsive to the needs of adolescents and support them to make safe and informed decisions about sex? Can the authors make recommendations pertaining to this issue.

See discussion on this in:

a. - Grossman et al. (2017). “We talked about sex.” “No, we didn’t”: Exploring adolescent and parent agreement about sexuality communication. American Journal of Sexuality Education. doi:10. 1080/15546128.2017.1372829

b. - Nilsson et al. (2020). Obstacles to intergenerational communication in care- givers’ narratives regarding young people’s sexual and reproductive health and lifestyle in rural South Africa. BMC Public Health, 20(791). doi:10.1186/s12889-020-08780-9

7. If socio-cultural norms impede open sexuality communication, is there a strategy for helping parents/elders to understand the implications of their attitudes and communication style on adolescent sexual decision making? Specific recommendations relating to this issue would be helpful. See discussion of this in:

a. - AVAC. (2018). Breaking the cycle of transmission: Increasing adoption of and adherence to effective HIV prevention among high-risk adolescent girls and young women.

b. - Duby et al. (2022) ‘I can't go to her when I have a problem’: sexuality communication between South African adolescent girls and young women and their mothers, SAHARA-J: Journal of Social Aspects of HIV/AIDS, 19:1, 8-21, https://doi.org/10.1080/17290376.2022.2060295

8. How can policy makers and programme designers ensure that interventions are designed and implemented in ways that are sensitive to local cultural and linguistic norms and beliefs? See the Nilson and Duby references cited above for discussion of this. Including recommendations to consider this would be beneficial.

We have address the issues raised in Nos. 3-8 above with an additional paragraph as follows:

Concerning parent-adolescent SRH communication, interventions that are likely to work are those that specifically train parents in soft communications skills such as talking less, listening more, being less judgmental, and asking more questions in interactions with adolescents about SRH [44]. Efforts should be directed at boosting the self-efficacy of parents (i.e. confidence in the ability to discuss sexual issues with their adolescent) by increasing their knowledge about adolescent SRH [45,46]. Also, interventions that will increase the motivation of parents to communicate with their adolescents, including both males and females, as well as on all appropriate SRH topics are likely to be effective [47]. An important consideration for program design is to create programs that target parent/caregiver-adolescent pairs as there is some evidence that they are effective [48]. The interventions that mediate the intergenerational gaps between parents and adolescents by finding a common ground for them are likely to be effective interventions as such gaps have been implicated in poor parent-adolescent communications [49]. Some interventions may be presented via traditional and social media in order to bring them to scale. For example, the use of radio drama has a good history of success in social behavior communication change [50,51]. Socio-cultural norms impede open sexuality communication, and it is a major impedance to parent-adolescent communication about SRH. Yet, interventions must be sensitive to local cultural and linguistic norms and beliefs for them to gain any traction. Adopting co-creation approaches from the ground up with the people for whom the interventions are targeted, as we did in this study, can help to navigate the norms that are peculiar to different groups.

---

## [Decision Letter · Decision Letter 3]

1 Dec 2022

Combination prevention package of interventions for reducing vulnerability to HIV among adolescent girls and young women in Nigeria: An action research

PONE-D-21-01166R3

Dear Dr. Arije,

We’re pleased to inform you that your manuscript has been judged scientifically suitable for publication and will be formally accepted for publication once it meets all outstanding technical requirements.

Kind regards,

Tai-Heng Chen, M.D.

Academic Editor

PLOS ONE

Reviewers' comments:

Reviewer's Responses to Questions

**Comments to the Author**

1. If the authors have adequately addressed your comments raised in a previous round of review and you feel that this manuscript is now acceptable for publication, you may indicate that here to bypass the “Comments to the Author” section, enter your conflict of interest statement in the “Confidential to Editor” section, and submit your "Accept" recommendation.

Reviewer #3: (No Response)

2. Is the manuscript technically sound, and do the data support the conclusions?

Reviewer #3: Yes

3. Has the statistical analysis been performed appropriately and rigorously? 

Reviewer #3: N/A

4. Have the authors made all data underlying the findings in their manuscript fully available?

Reviewer #3: Yes

5. Is the manuscript presented in an intelligible fashion and written in standard English?

Reviewer #3: Yes

6. Review Comments to the Author

Reviewer #3: This manuscript is much improved, and the authors have responded to most of the previous reviewer comments.

The new paragraph added relating to parent-adolescent SRH communication interventions does address some of my previous suggestions. However it was unclear which of the suggested references had been added and cited (this was unclear in the tracked changes). Additionally, there are some grammatical errors in the new paragraph - for example:

Line 576: "Socio-cultural norms impede open sexuality communication, and it is a major impedance to parent-adolescent communication about SRH."

- norms are plural, so "they" cannot be a singular impedance. This sentence should be revised and corrected.

The new paragraph should be throughly and carefully copy edited.

7. PLOS authors have the option to publish the peer review history of their article (what does this mean?). If published, this will include your full peer review and any attached files.

Reviewer #3: No

---

## [Editor Report · Acceptance letter]

27 Dec 2022

PONE-D-21-01166R3 

Combination prevention package of interventions for reducing vulnerability to HIV among adolescent girls and young women in Nigeria: An action research 

Dear Dr. Arije:

I'm pleased to inform you that your manuscript has been deemed suitable for publication in PLOS ONE. Congratulations! Your manuscript is now with our production department. 

Kind regards, 

on behalf of

Dr. Tai-Heng Chen 

Academic Editor

PLOS ONE